biomaterials/biomechanics/plant science

nutshell, secondary cell wall, interlocking, interface, microchemistry, tensile properties

**Author for correspondence:**
Notburga Gierlinger
e-mail: burgi.gierlinger@boku.ac.at

# Twist and lock: nutshell structures for high strength and energy absorption

Nannan Xiao, Martin Felhofer, Sebastian J. Antreich, Jessica C. Huss, Konrad Mayer, Adya Singh, Peter Bock and Notburga Gierlinger

Institute of Biophysics, University of Natural Resources and Life Sciences Vienna (BOKU), 1190 Vienna, Austria

 NX, 0000-0003-3350-7863; MF, 0000-0002-0701-3790;
SJA, 0000-0002-7030-3176; JCH, 0000-0003-3527-5154;
PB, 0000-0002-7032-9691; NG, 0000-0002-3699-9931

Nutshells achieve remarkable properties by optimizing structure and chemistry at different hierarchical levels. Probing nutshells from the cellular down to the nano- and molecular level by microchemical and nanomechanical imaging techniques reveals insights into nature's packing concepts. In walnut and pistachio shells, carbohydrate and lignin polymers assemble to form thick-walled puzzle cells, which interlock three-dimensionally and show high tissue strength. Pistachio additionally achieves high-energy absorption by numerous lobes interconnected via ball-joint-like structures. By contrast, the three times more lignified walnut shells show brittle LEGO-brick failure, often along the numerous pit channels. In both species, cell walls (CWs) show distinct lamellar structures. These lamellae involve a helicoidal arrangement of cellulose macrofibrils as a recurring motif. Between the two nutshell species, these lamellae show differences in thickness and pitch angle, which can explain the different mechanical properties on the nanolevel. Our in-depth study of the two nutshell tissues highlights the role of cell form and their interlocking as well as plant CW composition and structure for mechanical protection. Understanding these plant shell concepts might inspire biomimetic material developments as well as using walnut and pistachio shell waste as sustainable raw material in future applications.

## 1. Introduction

The complex hierarchical organization of biological shells results in exceptional mechanical performance, enabling the combination of strength and toughness [1–5]. Many biological systems incorporate minerals together with organic molecules to evolve

composites with unique structural capabilities and fracture resistance. For example, nacre consists of mineral platelets surrounded by an organic matrix, comparable to a brick- and mortar-like arrangement, which resists penetration, but also accommodates deformation. In arthropod exoskeletons (e.g. beetles and Crustaceae), helicoidal arrangements of chitin and collagen enhance the damage tolerance and energy absorption [6]. Some of these insights have inspired the development of synthetic structural materials [7–10]. In the plant kingdom also many hard shells have evolved, e.g. to protect seeds [11]. Protective shell tissues of plants consist of dead sclerenchyma cells, which usually have thick secondary cell walls (CWs) based on polysaccharides (cellulose and hemicelluloses) and lignin, an aromatic polymer. Lignification is the final step of secondary CW formation; monolignols are polymerized with oxidative enzymes, such as laccases and peroxidases [12]. Lignin is detected in the interface between sclerenchyma cells and on a lower hierarchical level—within the secondary CW as a matrix between cellulose micro- and macrofibrils and the associated hemicelluloses [13]. Sclerenchyma cells can have fibrous, isodiametric or irregular cell geometries. Recently, a new polylobate cell shape was identified in walnut and pistachio shells [14,15]. These three-dimensional puzzle sclereids achieve superior mechanical properties like higher tissue strength when compared with other nutshells based on fibrous and isodiametric sclereids [14,15]. Topological interlocking was highlighted as the main driver for high tensile strength in walnut and pistachio shells as compared with nutshells from other species. However, differences in the mechanical performance of the two puzzle cell species, walnut and pistachio, are so far without explanation and insights into the chemistry and structure of the puzzle cells on the micro and nanoscale still missing.

In this work, we tackle the secrets behind the different mechanical behaviour of pistachio and walnut shells. For both species, we investigate thin-walled (porous) and thick-walled (dense) tissue to disclose the effect of density. We trace structural and chemical differences down to the micro, nano- and molecular levels by X-ray tomography, electron microscopy, Raman microscopy and atomic force microscopy (AFM) to finally explain different mechanical properties and fracture surfaces. How cell geometry, puzzle cell interlocking, density, CW chemistry and nanostructure affect the mechanical performance of plant shells advances our understanding of natural packing structures.

# 2. Material and methods

## 2.1. Materials and sample preparation

Pistachios (*Pistacia vera*) were collected in June and September 2019 from Kerman, Iran. Walnuts (*Juglans regia*) were harvested in July and October 2018 from 'BOKU horticulture Jedlersdorf' in Vienna, Austria. All samples were frozen right after sampling to retain their native state and minimize chemical changes. We investigated thin-walled (porous) and thick-walled (dense) tissues of both species: for thin-walled (porous) tissue the inner part of walnut and the young pistachio shell were studied, and for dense tissue the thick-walled walnut cells of the outer shell part and the mature pistachio cells.

## 2.2. Light microscopy on single cells

The whole mature pistachio shells and part of walnut mature shells were macerated following the protocol of Frey *et al*. [16] with reaction times adjusted to our samples. Delignified single cells of pistachio and walnut were stained using astra-blue. The images of stained cells were acquired using a Labophot-2 microscope (Nikon) under bright field illumination.

## 2.3. Scanning electron microscopy

The microstructure of fracture surfaces was observed with an Apreo SEM (Thermo Fisher Scientific, Waltham, Massachusetts). To obtain images with higher resolution, the fracture surfaces were sputter coated with a gold layer of around 10 nm to improve conductivity and imaged, using an acceleration voltage of 1.0 kV and a current of 6.3 pA. The scanning electron microscopy (SEM) images were analysed with the software ImageJ [17] to determine the cellulose macrofibril diameter sizes [18] on the 41 and 31 measurements of pistachio and walnut, respectively; then mean values and corresponding standard deviations were calculated.

## 2.4. Micro-computed tomography

For obtaining micro-computed tomography (µ-CT) scans, dry shell fragments were scanned in an RX solutions EasyTom 150/160 system (RX solutions) (operating voltage of 40 kV, 150 µA current). Reconstruction was performed with the software XAct2 (RX solutions), and the resulting picture stacks were segmented in the software Amira (Thermo Fisher Scientific, Waltham, Massachusetts). For this purpose, cell lumina of individual cells were selected first, then all selections were expanded simultaneously until touching each other along the edges (similar to a three-dimensional Voronoi tessellation). The three-dimensional reconstruction of the final selections (minus cell corners (CCs) and intercellular space) led to three-dimensional models of each single cell in the tissue. The skeleton (yielding information about the number of lobes), area and volume, as well as the number of contact areas of neighbouring cells was measured digitally for 50 segmented cells. The difference in the number of lobes between pistachio and walnut was assessed using a Mann–Whitney U test (at $\alpha = 0.05$).

## 2.5. Fourier-transform infrared spectroscopy and Raman spectroscopy

Eight-micrometre-thick sections of pistachio shell (young and mature) were cut with a Cryostat Leica CM 3050 S (Leica) and that of the walnut shell with a rotary microtome Leica RM2235 (Leica), respectively. Fourier-transform infrared spectroscopy (FT-IR) spectra were acquired in the spectral range of 4000 to 800 cm$^{-1}$ with a resolution of 4 cm$^{-1}$ using the transmission mode of a Bruker Hyperion 2000 microscope. By mapping the sections of pistachio and walnut shells with a microscope aperture of $50 \times 50$ µm (acquisition field for one spectrum) multiple spectra (pistachio dense: 90 spectra; pistachio porous: 60; walnut dense: 90; walnut porous: 50) were obtained and averaged for each region of interest using Opus 7.5 software (Bruker Optik GmbH, Ettlingen, Germany). Spectra were baseline corrected and normalized at 1374 cm$^{-1}$ (assigned to cellulose and hemicellulose) to account for different tissue density (CW thickness). The peak area of the lignin bands was measured to compare the lignin content (relative to carbohydrates) of walnut and pistachio tissues.

Raman images (mappings) of cross-sections were acquired with a confocal Raman microscope (alpha300, WITec, Ulm, Germany) equipped with a piezo-scanner. The frequency-doubled Nd : YAG laser excitation ($\lambda = 532$ nm) was used in combination with a $100 \times$ oil-immersed (Nikon, NA = 1.25) microscope objective. For imaging, a laser power of 30 mW with an integration time of 0.1 s was used for pistachio shell and a laser power of 10 mW with an integration time of 0.01 s for walnut shell. The spectra were acquired using a CCD detector (DU401A-BV, Andor, UK) placed behind the spectrometer (UHTS 300; WITec). The software control FOUR 4.1 (WITec) was used for measurements and WITec Project FOUR 4.1 (WITec) and OPUS 7.5 software (Bruker) for spectrum processing. After applying cosmic ray spike removal, Raman chemical images were generated based on the integration of relevant wavenumber regions (e.g. lignin, carbohydrates). To calculate a lignin abundance ratio between compound middle lamella (CML) and CW, spatial average spectra for three individual images per category were obtained and peak area for the lignin bands measured in Opus 7.5 software.

## 2.6. Serial block face–scanning electron microscopy/transmission electron microscopy

The young (porous) walnut and pistachio shell were trimmed into $1 \times 1 \times 1$ mm cubes for serial block face (SBF)-SEM/transmission electron microscopy (TEM). Samples were immersed with a fixation solution (3% glutaraldehyde in 0.1 M sodium cacodylate buffer, pH 7.4) overnight at 4°C, post-fixed in 2% osmium tetroxide (OsO$_4$) and 0.2% ruthenium red in 0.15 M cacodylate buffer for 1 h at room temperature followed by five times washing in 0.15 M cacodylate buffer. Then, samples were immersed for 45 min in freshly prepared 1% thiocarbohydrazide solution followed by thrice washing in distilled water and a second post fix step in 2% OsO (1 h at room temperature) followed by four times washing. Afterwards, the samples were immersed into 0.5% uranyl acetate solution overnight (4°C) followed by five times washing and additionally immersed in Waltron's lead aspartate solution for 30 min (65°C) followed by five times washing. Graded series of ethanol concentrations of 30%, 50%, 75%, 90% and 100% were used for the dehydration process, and the ethanol excess was removed by rinsing twice with 100% acetone. Then, the samples were embedded in low-viscosity epoxy resin (LV-resin), starting incubation with 25% LV-resin in acetone overnight (4°C), followed by 4 h with 50% and 4 h with 75%. Then, the samples were incubated overnight in 100% LV-resin, followed by another round in 100% LV-resin (6 h at room temperature) The embedded samples were placed in a silicon mould and polymerization was performed at 65°C for 48 h. For three-dimensional imaging, embedded blocks were trimmed and mounted on the

volumescope microtome in the Apreo SEM (Thermo Fisher Scientific, Waltham, Massachusetts) and picture stacks of the tissues were acquired. Picture stacks were aligned in ImageJ and several cells were three-dimensionally reconstructed using the software Amira. For TEM, thin sections of 80 nm were obtained from the embedded blocks using an Ultracut UC7 microtome (Leica Microsystems, Vienna, Austria) and stained with potassium permanganate ($KMnO_4$). Images were obtained in a Tecnai T20 TEM (FEI, Hillsboro, Oregon). The software ImageJ was used to measure the laminate distance in the TEM pictures by analysing profiles along with the CW. At least 20 measurements were obtained for walnut and pistachio, and average values as well as standard deviations were reported.

## 2.7. Atomic force microscopy

AFM measurements were performed using the Witec atomic force microscope alpha300 in the digital pulsed force mode. The dense walnut and pistachio shell blocks were faced with an Ultracut UC7 microtome (Leica), cleaned with nitrogen gas and then placed on a three-dimensional-printed holder. Forces (nN scale) were applied with a tetrahedral silicon tip (ArrowTM, Nanoworld, Switzerland) with 10 nm radius of curvature and a spring constant of $2.8 \, N \, m^{-1}$. Details of data analysis and image reconstruction were described by Felhofer *et al*. [19].

## 2.8. Tensile test

The shell segments for tensile test were trimmed to rectangular bar specimens (thickness approx. 0.8–1.2 mm, width approx. 1–2 mm and length approx. 0.8–1 cm) with a Cryostat Leica CM 3050 S (Leica). Tensile properties of the dense walnut shell as well as the porous and dense pistachio shell were measured using a material testing machine (zwickiLine Z2.5, Zwick-Roell) with a 1 kN load cell (Zwick-Roell) and a video extensometer (videoXtens, Zwick-Roell). All the nuts specimen preparation details were described by Huss *et al*. [15]. The tested nutshells include dense pistachio ($n = 20$ including 10 samples from [15], from Iran, 2019); porous pistachio ($n = 10$, from Iran, 2019) and walnut ($n = 10$, from Austria, 2018). Differences of measured elongation at rupture, tensile strength, energy absorption as well as Young's modulus among the groups (walnut shell as well as the porous and dense pistachio) were assessed using Kruskal–Wallis tests with subsequent pairwise post hoc tests (Mann–Whitney U test using Bonferroni correction to adjust for multiple comparisons, at $\alpha = 0.05$ for all hypothesis tests). The fractured surface of the samples was observed using Apreo-SEM, lower magnification with a voltage of 1.0 kV and 50 pA current and higher magnification with an acceleration voltage of 2.6 kV and 0.1 nA current.

# 3. Results

## 3.1. Three-dimensional puzzle cells: lobes increase surface area

The polylobate cell shape is best visible in young (porous) pistachio shells (figure 1*a*), while in the mature pistachio shell, the tissue is denser and the shape more difficult to discern (figure 1*b*). In the entire pistachio shell, all cells have a rather uniform wall thickness (figure 1*b*), while walnut shells have even in the mature state thin-walled cells (porous tissue) towards the seed (inner part) (figure 1*c*). Disintegrating the tissues into single three-dimensional puzzle cells opens the view on the cell shape and suggests pistachio cells to be more lobed than the walnut cells (figure 1*a–c*; electronic supplementary material, figure S1). X-ray tomography confirms approximately thrice more lobes in pistachio puzzle cells ($37 \pm 11$) compared with walnut cells ($12 \pm 5$) (figure 1*d*). A common principle is that both species interlock one puzzle cell with 14 neighbouring cells on average (figure 1*e*). So, the nearest geometrical configuration is the tetrakaidecahedron with 14 faces, which is one of the most efficiently packing geometries in space [20]. Compared with tetrakaidecahedron, the walnut puzzle cells increase the surface area per unit volume up to 30–40% [14] and the pistachio puzzle cells up to 60–80% due to more lobes (figure 1*f*).

## 3.2. Molecules in context with microstructure: lignification of the cell interface and the cell wall

As interfacial glue between the cells, pectin plays a major role in growing cells with primary CWs [21], whereas lignin is detected between sclerenchyma cells after the final cell size and shape is reached [22].

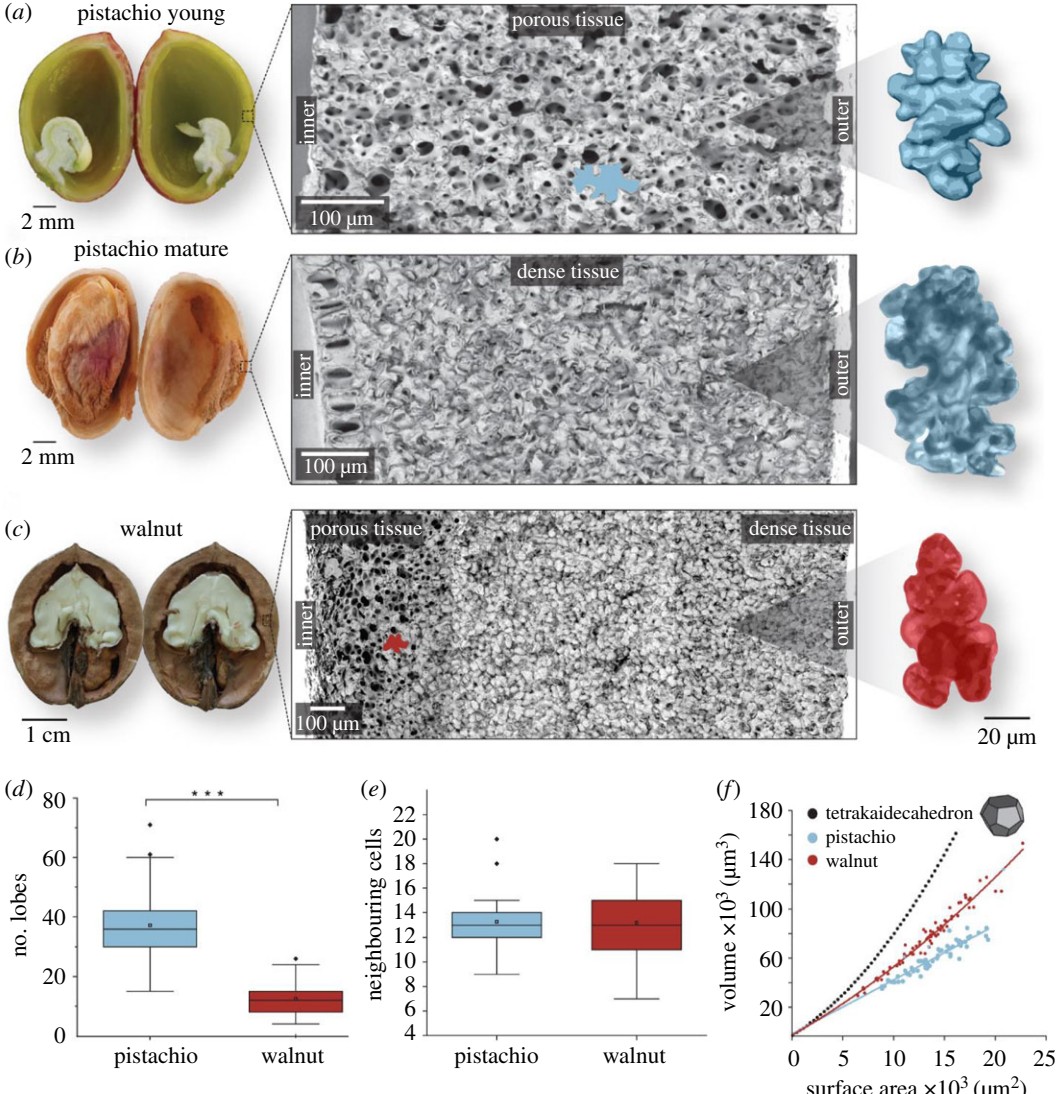

**Figure 1.** Structural characterization of pistachio and walnut shells. (a–c) From left to right: macroscopic images of opened shells, cross-sectional SEM images of the shell (insert outline of the cell) and a single-cell representation based on X-ray nanotomography for young (porous) pistachio and respective light microscopy for the dense (mature) pistachio and dense walnut shell. Colours were changed for better comparison (original images in electronic supplementary material, figure S1). (d) Comparison of the number of lobes per cell of pistachio ($n = 59$) and walnut cells ($n = 77$) and (e) of the number of neighbouring cells of pistachio and walnut. (f) The higher number of lobes leads to a higher surface area at a given volume for pistachio as compared with walnut.

Based on infrared spectra (the 1504 cm$^{-1}$ band assigned to aromatic ring vibrations [22]), approximately 40% less lignin is found in pistachio shells compared with walnut shells (figure 2a). As lignin is present in the interface between the cells as well as within the CW [22], a change in lignin content can affect the final properties of tissues via interface and/or CW composition. To quantify the lignin abundance in the CW in comparison with the interfaces between the cells (CML and CC), Raman imaging of lignin was performed (figure 2b,d). Similar to wood [24] lignin in the pistachio shell tissue is more abundant between the cells in CML and CC (figure 2b). In the porous pistachio shell tissue, lignin concentration is about four times higher in the interface compared with the CW. The latter becomes increasingly more lignified during CW thickening, causing the ratio to drop to 2 in the dense cells (figure 2b,c; electronic supplementary material, figures S2 and S3). In the walnut shells, the lignin content in the CW is higher than in pistachio and the difference to the interfaces (CML) is less pronounced (1.5 times for dense part and 1.7 times for porous part of the shell) (figure 2d,e; electronic supplementary material, figure S3).

**Figure 2.** Chemical analysis by IR and Raman microspectroscopy of pistachio and walnut shells. (*a*) FT-IR spectra from porous resp. dense pistachio shells and porous resp. dense part of walnut shells after baseline-correction and normalization at 1374 cm$^{-1}$ (represents cellulose and hemicelluloses) [23] show that the intensity of the lignin peak in walnut (0.7 rel. intensity) is about three times higher than in the dense pistachio shell (0.2 rel. intensity) in relation to the carbohydrates. (*b–e*) High-resolution Raman images display the lignin distribution based on the integral of the marker band at 1600 cm$^{-1}$ in porous and dense pistachio shells (*b*) and porous and dense part of the walnut shells (*d*). Lignin bands area ratio between CML and CW in porous and dense pistachio shell (*c*) and the porous and dense part of the walnut shell (*e*), based on band integration of the peaks from 1558 to 1698 cm$^{-1}$ in the Raman spectra.

## 3.3. Cell wall on the nanolevel: helicoidal cellulose sheets (Bouligand structure) assembled in laminated structures

In a recent study, the nanochemical composition of the wood CW was retrieved by AFM using the adhesion values based on the pulsed force mode [19]. Applied to dense walnut and dense pistachio tissues, nanodomains are visualized based on adhesion values and a higher lignin content in walnut shells was confirmed on the nanolevel (electronic supplementary material, figure S4).

But how about the structure of the cellulose macrofibrils as important drivers of mechanical properties of the plant CW? The AFM topography images show a periodically alternating layering for pistachio and walnut shells (figure 3*a*). The corresponding height profiles reveal that the lamellae are thinner in pistachio CWs (138.9 ± 8.2 nm) than those of walnut (228.2 ± 24.2 nm) (figure 3*b*; electronic supplementary material, table S1). Within the plant CW, the cellulose microfibrils serve as stiff elements with a longitudinal modulus of elasticity of around 138 GPa in the crystalline regions [25], embedded in the soft matrix of hemicellulose and lignin polymers [26]. However, the measured modulus of elasticity depends on the angle of the cellulose microfibrils relative to the indentation direction, as shown for CWs of wood [27]. Thus, the observed changes in AFM stiffness reflect changes in the orientation of the cellulose macrofibrils within puzzle CWs (electronic supplementary material, figure S5). The images obtained by TEM on cross-sections confirm the lamellar structure of the puzzle CWs (figure 3*c*; electronic supplementary material, table S1). Furthermore, the regular arced patterns could be identified in both TEM images, as well as porous pistachio TEM image (electronic supplementary material, figure S6), and correspond to a helicoidal organization (Bouligand structure), i.e. each stack of cellulose macrofibrils (laminae) is parallel to each other with a stepwise rotation. Based on high-resolution SEM images (figure 3*d*), the average diameter of the macrofibrils ($d_f$) is around 25 nm for both pistachio and walnut (electronic supplementary material, table S1). In both models, we assume no additional space between macrofibrils. Based on calculation, around six laminae stack to make a single wall lamella (pitch) in pistachio and 10 laminae in walnut (figure 3*e*; electronic supplementary material, note S1 and table S1). A lower number of laminae per pitch in pistachio leads to a higher calculated pitch angle ($\gamma = 32°$) compared with an angle of 18° in walnut (figure 3*e*; electronic supplementary material, table S1). The pitch angle can modulate the mechanical

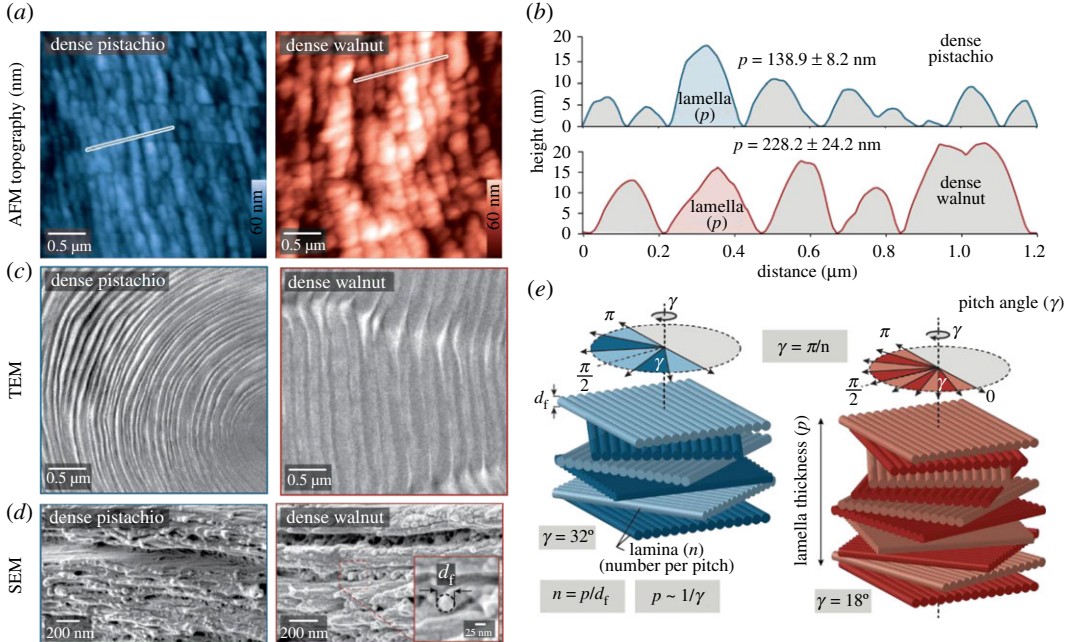

**Figure 3.** Nanostructure of the shell of dense pistachio (blue) and dense walnut (red). (*a*) AFM topography images. (*b*) Corresponding height profiles (along the lines in *a*), showing single CW layers with lamellae of different thicknesses (*p*). (*c*) TEM images of the CW. (*d*) High-resolution SEM images of CW layers. (*e*) Model of the helicoidal CW architecture: the laminae (*n* is the number of laminae per pitch) rotate with the pitch angle (*γ*), for every 180° (*π*) a lamella or pitch (with the thickness *p*) is observed. Based on calculations, pistachio shows a larger pitch angle (32°) between the laminae when compared with walnut (18°), involving thinner lamellae, because the fibril diameter ($d_f$) is similar for both. The lamella thickness (*p*) is inversely proportional to the pitch angle ($p \sim 1/\gamma$).

properties [28]. The Young's modulus of the pistachio CW on the nanolevel is indeed higher than that of walnut (electronic supplementary material, figure S7 and table S1).

## 3.4. Tensile testing and fracture surfaces of three-dimensional puzzle cell tissues

Micromechanical tensile testing of walnut shell specimens shows stress–strain curves typical for brittle materials, while the pistachio shells are strained up to 8% (figure 4*a*; electronic supplementary material, figure S8a and table S2). This remarkable elongation is also observed in porous pistachio shells with lower strength (figure 4*a*; electronic supplementary material, figure S8 and table S2). The lower tensile strength (electronic supplementary material, figure S8b) and modulus of elasticity reflects the thinner secondary CWs in the porous young tissue (figure 4*b*). The Young's modulus of dense pistachio (*ca* 2.6 GPa) and walnut shells (*ca* 2.8 GPa) as well as the tensile strength are similarly high (figure 4*b*; electronic supplementary material, figure S8b and table S2). However, as a result of the increased strain in dense pistachio (figure 4*a*), also the energy absorption is larger for pistachio (2.6 MJ m$^{-3}$) than for walnut (1.2 MJ m$^{-3}$) (figure 4*c*; electronic supplementary material, table S2).

The fracture surfaces under the SEM open the view on relevant structural features on the cellular level (figure 4*d*,*e*; electronic supplementary material, figure S9). The thin CWs in the porous part of walnut and in porous pistachio shell fracture across and thus display the inner surface (porous part of walnut in figure 4*d* and porous pistachio in figure 4*e*). The thick-walled dense part of walnut shells shows partly intact sclerenchyma cells, interfaces (CML) and pits, figure 4*d*), and also a few ruptured lobes exposing the lamellar structure (figure 4*d*). In addition, intercellular spaces were found in the walnut shell, in the dense as well as in the porous part (figure 4*f*). In pistachio, not only the thin-walled porous three-dimensional puzzle cells fracture across the CW, but also the thick-walled denser cells (figure 4*e*). Compared with walnut, more lobes with ball-joint-like connections are revealed in pistachio, which suggests stronger topological interlocking (figure 4*g*). In walnut shell tissues, more pits perforate the secondary CWs and fractures are observed aside pits (figure 4*d*; electronic

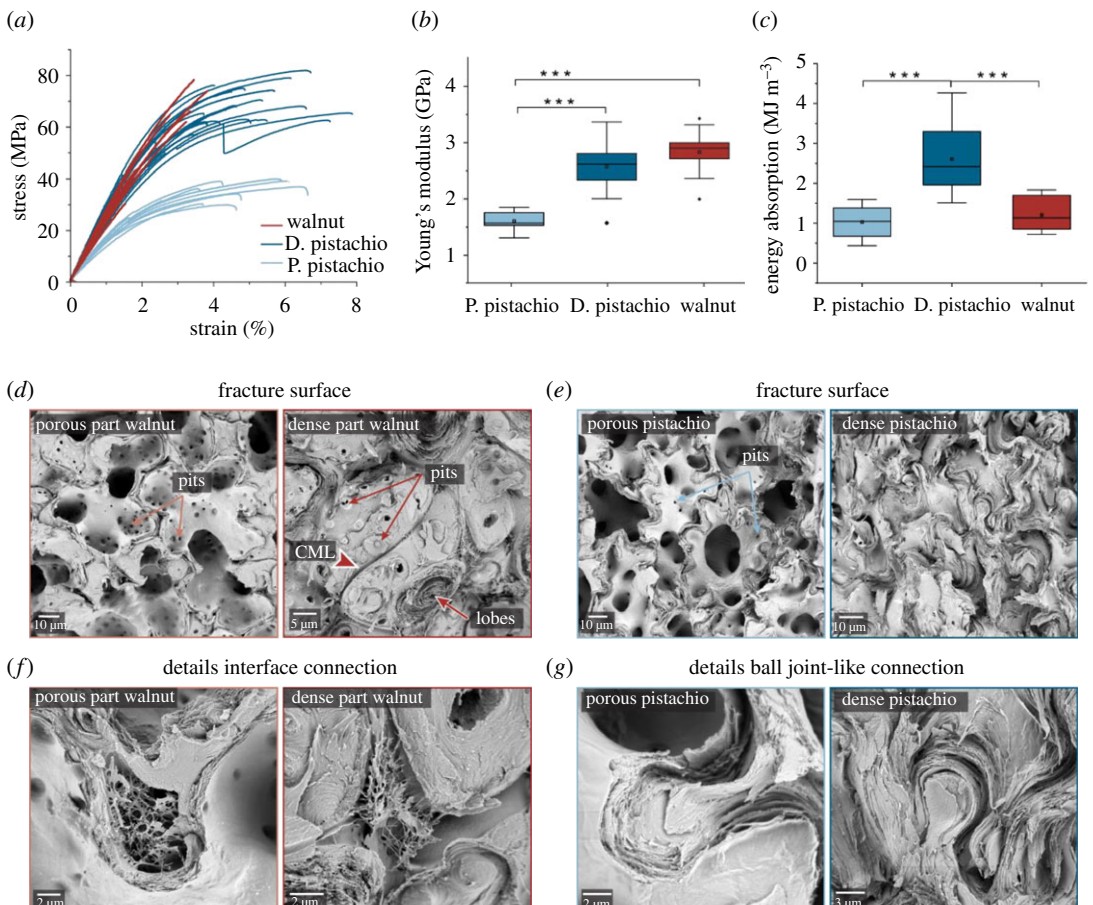

**Figure 4.** Mechanical characterization and microstructure of pistachio and walnut shells. (*a*) Comparison of the tensile stress–strain curves of dense pistachio shells (sampled September, $n = 20$; 10 taken from Huss *et al.* [15]), porous pistachio shells (sampled June, $n = 10$) and walnut shells (sampled September, $n = 10$). (*b*) Tensile modulus and (*c*) energy absorption, obtained by integration of the stress–strain curves. (*d*) SEM images of the fracture surfaces of the porous and dense walnut shells (*e*) SEM images of the fracture surfaces of the porous and dense pistachio shells. (*f*) Details of the fracture surface of walnut show the intercellular space between cells. (*g*) Details of the fracture surface of pistachio reveal a ball-joint-like connection.

supplementary material, figure S10a,b). During secondary CW formation, intramural pits are kept open to maintain the function of the plasmodesmata [29].

# 4. Discussion

## 4.1. Three-dimensional puzzle shells: strength and energy absorption by optimization at different length scales

To combine strength and toughness is a big challenge in material science and engineering [30]. Stiff and strong building blocks have to be arranged to interact through energy-dissipative interfaces [9]. The stiff and strong building blocks in dense pistachio and walnut shells on the cellular level are thick-walled, lignified three-dimensional puzzle cells, which result in a very dense tissue (figures 1 and 2) with increased interfaces (cell contact area) and superior mechanical properties compared with other nutshells due to interlocking [15]. Between these two puzzle cell tissues, remarkable structural and chemical differences have been elucidated at the micro and nanolevel and are now discussed with regard to the different mechanical performance, especially in terms of energy absorption.

Based on the results (figures 1–4), we summarized and schematized the most important structure and fracture characteristics of walnut (figure 5*a–d*) and pistachio shells (figure 5*e,f*). In dense walnut tissue, the three-dimensional puzzle cells separated mainly along the CML (separation of cells) or the lamellae of the CW, resulting in separated cells and a few ruptured lobes (figure 5*a*), while the porous walnut tissue

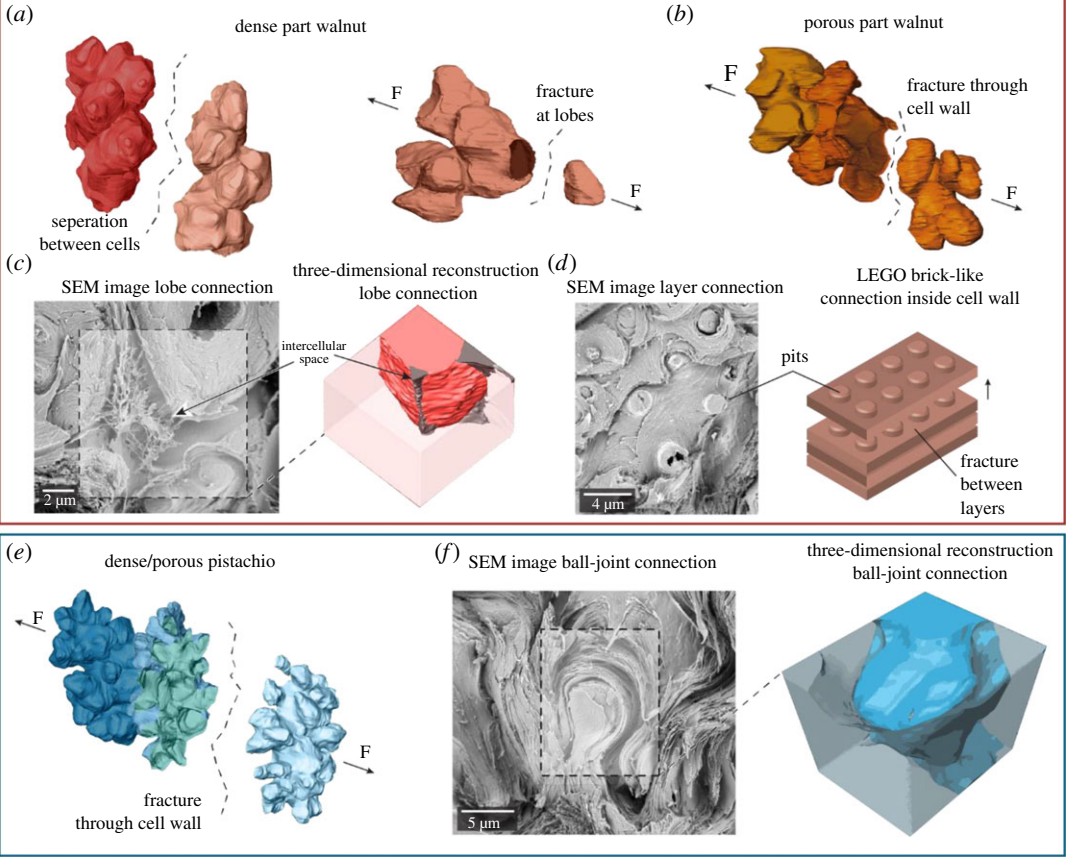

**Figure 5.** Cellular three-dimensional segmentation models, fracture characteristics and schematics in walnut and pistachio shells. (*a*) In dense walnut shells, whole cells are pulled apart at the interface or lamellae and lobes are ruptured. (*b*) In porous walnut shells, the thin CW fractures across the cell (*c*) SEM image and three-dimensional segmentation of lobes connection in walnut obtained by SBF-SEM. (*d*) The sticking-out pits remind one of a LEGO-brick-like connection within the CW and single lamellae (schematic drawing). (*e*) In dense/porous pistachio, the fracture surface reveals the rupture within the CW. (*f*) SEM image and three-dimensional segmentation of ball-joint connection in pistachio obtained by SBF-SEM.

ruptured mainly through the thin CWs (figure 5*b*). Intercellular spaces between the lobes (figure 5*c*) as well as the numerous pits passing through the cell walls might be partly responsible for the more brittle fracture behaviour (figure 4*a*). Around pits the lamellae and cellulose macrofibrils divert their orientation (electronic supplementary material, figure S10*c*) and pits might therefore influence crack propagation. This was deduced from fracture surfaces with layer-by-layer fracture around the pits and their sticking-out CWs reminding one of LEGO-bricks (figure 5*d*). Pit channels go from cell to cell and are needed to maximize the contact area between symplast and apoplast to facilitate transport and diffusion processes for CW construction, including cellulose deposition and lignification [31]. The numerous pit channels in walnut might thus be linked with the higher lignin content proven by FT-IR spectroscopy (figure 2*a*). The higher lignin content in walnut is in agreement with Landucci *et al.*, who reported a total lignin content of *ca* 32% in walnut shells and *ca* 17% in pistachio shells [32]. Lignin is known to play a role in the mechanical properties of plant CWs, e.g. in poplar wood axial stiffness was higher in wild-types than in mutants with reduced lignification [33]. On the other hand, less lignin gives a weak interface, which facilitates high-energy absorption (toughness) [34], as observed in pistachio (figure 4*c*). Lignin is found on the microlevel in the interface between the cells and on the nanolevel between the cellulose fibrils within the CW (figure 2*b–e*). While the pistachio CW had a low lignin content in the CW (figure 2*b,c*), similar high lignin contents were detected in the CW and interface of the walnut shell tissue (figure 2*d,e*). Less lignin laid down between the cellulose macrofibrils could be one reason behind the detected thinner lamellae in pistachio. More stiff cellulose macrofibrils and their helicoidal arrangement (figure 3) provide strength and energy absorption, as lamellae dissipate energy when cracks propagate through [4,35,36]. The pitch angle of the helicoid motif was shown to tune the mechanics of helicoidal architecture [28]. As shown by

Naleway *et al*. [37], the macrofibrils are only capable of resisting load in the direction of tension. Therefore, a higher pitch angle in the pistachio shell will give more force scattering and higher energy dissipation for defined numbers of laminae.

We can only speculate about why both species evolved shells with such different lignin content. From a biological point of view, the saving of lignin in pistachio might be a result of better protection by an inner cuticular layer (electronic supplementary material, figure S11) and an outer husk surrounding the shell, even after maturity. Lignin usually replaces water in plant tissues and adds hydrophobicity and thereby increases biotic and abiotic resistance [38]. Or could pistachio save the lignin due to the fact that the shell tissue evolved as the master of geometrical cell interlocking? Besides the above-discussed thinner lamellae and higher pitch angle (figure 3), we found two geometrical features optimizing the remarkable mechanical protection in the pistachio shell tissue. First, more lobes (approximately three times more than walnut) increase the surface area more than 30% in pistachio (figures 1 and 5*e*). Such a large surface area is beneficial for linking and bonding the cells, physical adhesion at the interface, stress transfer and ultimate mechanical properties of composites [39]. Second, ball-joint-structures optimize interlocking of the three-dimensional puzzle cells (figures 4*g* and 5*f*). They help to explain why both young thin- and thick-walled pistachio shell tissue showed a similar strain (figure 4*a*) and fracture through the CW (figure 5*c*). Interdigitated sutures together with laminated microstructures have recently been shown to be the most important toughening mechanisms in the elytra of the diabolical ironclad beetle [40]. In seahorse armour, interlocking joints of the bony plates achieve large deformations [41,42]. Highly deformable bodies are essential in soft robots, prosthetics and orthotics and have been realized by bioinspired functionally graded three-dimensional-printed ball and socket joints [43,44].

## 4.2. Nutshells: valuable waste for new material developments and applications

Knowledge about the lignin content and its distribution in nutshells is not only of importance to understand its role in plant shell packaging, but is also of high importance considering a potential technical use of lignified nutshell waste. Disintegration of the shells by delignification results in single polylobate sclereid cells with high surface area (figure 1*a–c*; electronic supplementary material, figure S1). This unique and uniform cell shape in nut shells could inspire new concepts of topological interlocking, which would help to create new architecture materials. Recently, delignification and disintegration of wood followed by functionalization and/or densification has led to new horizons in the development of functional advanced materials [45–47]. Likewise, there is high potential in the use of polylobate nutshell cells. In future work, we might use biochemical adhesives embedding the polylobate cells to create large-scale and fracture-resistant materials. Furthermore, delignified and modified walnut single cells with multiple interconnected pits, but still stable due to a thick secondary CW, could be the basis material to find improved strategies for wastewater treatment compared with the raw shell powder [48].

The unique hierarchical microstructures of plant fibres (ramie, jute, kenaf, etc.) are given special attention for reinforced sustainable composites used in automotive, infrastructure, sports and even aerospace industry [49]. Also natural and engineered wood, microscale holocellulose fibres and nanoscale fibrils show high potential for new sustainable multifunctional structural materials [47,50]—boosted through a better understanding of structure–property–function relationships by improved characterization methods [45,51]. Helicoidally organized cellulose macrofibrils are able to build strong plant surfaces [52], but also induce colour and iridescence [53]. Cholesteric-like organization of cellulose structures of plants serve currently as an inspiration for the production of next-generation soft interactive materials [54].

## 5. Conclusion

In this work, we identified the ingenious cell shape and arrangement of biomolecules of walnut and pistachio shells to explain the remarkable mechanical performance. Pistachio shells turned out as the master of three-dimensional puzzle cell interlocking—ball-joint-like structures explain the high-energy absorption and deformability. So, not only the animal kingdom and bioinspired robotic bodies use these concepts, but also pistachios. In contrast, the highly lignified walnut shells crack with brittle LEGO-brick failure along the interface (middle lamellae), along the numerous pit channels and laminated CW structures. The helicoidal architecture of cellulose explains in general the exceptional

mechanical properties of the sclerenchyma cells of nutshells and reminds one of similar laminated crack propagation structures in animals, e.g. beetles. With the pitch angle mechanical properties can be tuned: a higher pitch angle in pistachio shell together with lower lignin content might additionally contribute to the observed higher energy absorption. Understanding nutshell structure and composition at different hierarchical levels can inspire biomimetic material developments and pave the way for new sustainable materials based on nutshell waste. The uniform isotropic cell type throughout the whole shell and the tremendous surface area found in walnut and pistachio cells might outperform anisotropic fibres in some industrial applications. The two varieties of three-dimensional puzzle cells come along with different micro- and nanostructure and chemistry and might thus be chosen for specific applications, accordingly.

Data accessibility. The data that support the findings of this study are available from the corresponding author upon reasonable request.

Authors' contributions. Conceptualization was done by N.G. and N.X.; data acquisition and analysis were done by N.X., S.J.A., J.C.H., M.F., A.S. and P.B.; funding and resources were done by N.G.; original draft was done by N.X., P.B. and M.F.; interpretation and review were done by all authors.

Competing interests. The authors declare no competing interests.

Funding. This work is supported by the European Research Council (ERC) under the European Union's Horizon 2020 research and innovation programme grant agreement no. 681885.

Acknowledgements. We thank the whole bionami research group for helpful comments (www.bionami.at), as well as Alemeh Karami and Karl Refenner (BOKU Versuchszentrum) for providing nut samples.

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
