## [Peer Review File · Royal Society Open Science]

Review History

RSOS-210399.R0 (Original submission)

Review form: Reviewer 1

Is the manuscript scientifically sound in its present form?

Yes

Are the interpretations and conclusions justified by the results?

No

Is the language acceptable?

Yes

Do you have any ethical concerns with this paper?

No

Have you any concerns about statistical analyses in this paper?

No

Recommendation?

Major revision is needed (please make suggestions in comments)

Comments to the Author(s)

The article presents the structure and mechanical characteristics of nutshells, comparing pistachio and walnut. The interpretation of the mechanical analysis is based on the structural and compositional analysis at the micro- to nano- scales. This innovative work is presented in very clear figures. In some cases, interpretation is entangled to the results that makes it hard to distinguish measurement data from implications. This is especially present when mechanical data is interpreted based on microscopy of fractured surfaces- Fig. 4. A clear separation between collected and inferred data will strengthen this work significantly.

Specific comments:

Abstract- I am not sure to follow the connection between pits and lignification.

Line 72- Please explain why frozen samples are the best to preserve the structure, as freezing may damage the sample.

Line 213- what does the normalization peak at 1374 cm^{-1} represent?

Line 214- The peak ratio is not reported- we (readers) do not see the 3X factor.

Line 217- some word rearrangement is needed- either explain the abbreviation or remove parenthesis

Line 219- Please add "in the Raman spectra" or something in this line.

Line 254- Figure 3 does not have mechanical information, but only topography, measured by SEM and AFM. Based on topography you suggest a helicoidal structure. Panel c is very confusing because it is presented as if the SEM and TEM images are of similar scales and similar structures, but the TEM features are about 3-5 times smaller. Please refer to this in the text and explain what the (SEM) bigger features are. I suggest moving the TEM panels together with the high resolution SEM in panel d.

Line 268- why DESPITE the lower strength? Low strength may appear together with high or low strain. The high strain in both dense and porous tissues is very interesting and worthy of a separate discussion.

Line 270- another important variation is the increased lignification. Is the higher strength due to wall density or wall lignin content? The place to discuss this is the discussion. Appearing in the results part, this confuses the reader.

Line 275- the striking variation between walnut and pistachio is the high strain. This results in a higher energy.

Line 282- "Numerous pits..." Are there more pits in the walnut sclerenchyma as compared to pistachio? Are there more pits in the dense as compared to porous walnut shell? Can you measure the pit density?

Line 288- "influence crack propagation" - you have no evidence for this and therefore this is an interpretation to the data- please move to discussion. The nice LEGO connection is not in place either. I suggest separation of Fig. 4e and the inset in Fig. 4d to a fifth figure, which is hypothetical and presented in the discussion. The sentence at line 289-290 is not clear, and I suspect that it is too hypothetical, and not based on measurements.

Figure 4- Missing parenthesis after ref. [15]. Typo on the title in panel d- missing dense part walnut. This panel is not very clear- I suggest moving the arrowhead to a more central position in the image. As it is it seems to point to the figure edge. This panel has arrows pointing to the LEGO motifs that may be confused with the arrow pointing to the missing lobe.

Discussion-

In general, the discussion is not focused, and thus the reader is lost. I suggest to concise the text and add a figure explaining your hypotheses regarding the mechanical model, highlighting the difference between the four tissues you analyzed.

Line 319-on- Please move the paragraph to active rather than passive voice.

Line 328- the sentence starting at the end of this line is unclear and too long. In addition, a reference is needed for the comment on “remarkable mechanical performance”.

Line 330- seA shells (?)

Line 350- I am not sure about this statement- pits are for cell-cell connections through plasmodesmata, not cell-extracellular space connection.

Line 351- transport of monolignol to the apoplast is possibly done by diffusion, specialized transporters, or exocytosis, but not through pits.

Line 353-4 - please cite a reference for this claim

Review form: Reviewer 2

Is the manuscript scientifically sound in its present form?

Yes

Are the interpretations and conclusions justified by the results?

Yes

Is the language acceptable?

Yes

Do you have any ethical concerns with this paper?

No

Have you any concerns about statistical analyses in this paper?

No

Recommendation?

Accept with minor revision (please list in comments)

Comments to the Author(s)

In their manuscript “Twist and Lock nut shell structures for high strength and energy absorption” Xiao et al report the structural, chemical and mechanical characterisation of walnut and pistachio shells and provide novel insight in the underlying structure/chemistry – function relationships.

In general, the manuscript is of high quality, including very interesting analytical approaches and results; however, before suitable for publication the following points should be addressed:

- 1) The authors report a different lignin content for the two different nut samples (based on IR/Raman measurements, semi-quantitative methods, and literature values) is it possible for the authors to provide wet chemistry results as quantitative analysis for the lignin content?
- 2) The authors analyse fracture surfaces (after mechanical testing) to elucidate structural features – how to ensure that the fractured surfaces are representative for the native state structure; for example in the case of wood cells (cell walls) fractured surface reveal a different structure, compared to sample surfaces prepared by other techniques and the native state;
- 3) Page 5 line 92-ff: “cell lumina were selected and expanded simultaneously until touching each other along the cell wall” –I have to admit that I cannot entirely follow the process described here – the procedure should be described a bit more in detail.
- 4) Page 6 line 103: IR spectra: the authors mention that spatial average spectra were utilized; what exactly do the authors mean with that – was a mapping performed? Or multiple IR spectra averaged?

5) AFM - Adhesion images (SI): it would be very helpful if the authors could explain more in detail how the data for the lignin/ carbohydrates adhesion values were fitted-even though the authors refer to a previous work; it would be interesting to perform the same analysis with the delignified samples; Do the results change accordingly when the porous pistachio (less lignin) is analyzed?

Typically, the topography is influenced by the orientation of the cellulose fibrils - how is this accounted for?

6) Raman Analysis: the authors report changing lignin content between CW and CML/CC with different ratios between pistachio and walnut; lignin peaks, eg. the utilized bands at 1658 and 1600 1/cm, (intensity) can also be influenced by a different lignin structure/not only amount or by aromatic extractives (common for walnut)-can the authors comment on that? In particular, considering that in the provided images in Figure 2b it looks like there is quite some underlying fluorescence (also reported by the authors in a previous paper for walnut shells+S10); Are the averages taken from different cell walls;

7) Mechanical properties: in the case of the walnut the analyzed sample comprise dense and porous parts, whereas for pistachio dense and porous parts are separated? Is this of influence for the results?

8) Figure 4 d: I assume the right part should be the dense part;

9) Perspectives of using the waste shell waste for materials: can the authors be a bit more specific how the waste could be utilized in composite materials and to profit from the nutshell intrinsic structure.

Decision letter (RSOS-210399.R0)

Dear Dr Gierlinger

The Editors assigned to your paper RSOS-210399 "Twist and lock: nut shell structures for high strength and energy absorption" have now received comments from reviewers and would like you to revise the paper in accordance with the reviewer comments and any comments from the Editors. Please note this decision does not guarantee eventual acceptance.

Please submit your revised manuscript and required files (see below) no later than 21 days from today's (ie 18-May-2021) date. Note: the ScholarOne system will 'lock' if submission of the revision is attempted 21 or more days after the deadline. If you do not think you will be able to meet this deadline please contact the editorial office immediately.

on behalf of Professor Guy Genin (Associate Editor) and Pietro Cicuta (Subject Editor)
openscience@royalsociety.org

Associate Editor Comments to Author (Professor Guy Genin):

Associate Editor: 1

Comments to the Author:

Thank you for submitting this very interesting paper to RSOS. Like me, the reviewers find your results and ideas compelling. The reviewers have some constructive criticism about how your paper might be improved, and I encourage you to follow their advice in the revised manuscript.

Associate Editor: 2

Comments to the Author:

(There are no comments.)

Reviewer comments to Author:

Reviewer: 1

Comments to the Author(s)

The article presents the structure and mechanical characteristics of nutshells, comparing pistachio and walnut. The interpretation of the mechanical analysis is based on the structural and compositional analysis at the micro- to nano- scales. This innovative work is presented in very clear figures. In some cases, interpretation is entangled to the results that makes it hard to distinguish measurement data from implications. This is especially present when mechanical data is interpreted based on microscopy of fractured surfaces- Fig. 4. A clear separation between collected and inferred data will strengthen this work significantly.

Specific comments:

Abstract- I am not sure to follow the connection between pits and lignification.

Line 72- Please explain why frozen samples are the best to preserve the structure, as freezing may damage the sample.

Line 213- what does the normalization peak at 1374 cm⁻¹ represent?

Line 214- The peak ratio is not reported- we (readers) do not see the 3X factor.

Line 217- some word rearrangement is needed- either explain the abbreviation or remove parenthesis

Line 219- Please add "in the Raman spectra" or something in this line.

Line 254- Figure 3 does not have mechanical information, but only topography, measured by SEM and AFM. Based on topography you suggest a helicoidal structure. Panel c is very confusing because it is presented as if the SEM and TEM images are of similar scales and similar structures, but the TEM features are about 3-5 times smaller. Please refer to this in the text and explain what

the (SEM) bigger features are. I suggest moving the TEM panels together with the high resolution SEM in panel d.

Line 268- why DESPITE the lower strength? Low strength may appear together with high or low strain. The high strain in both dense and porous tissues is very interesting and worthy of a separate discussion.

Line 270- another important variation is the increased lignification. Is the higher strength due to wall density or wall lignin content? The place to discuss this is the discussion. Appearing in the results part, this confuses the reader.

Line 275- the striking variation between walnut and pistachio is the high strain. This results in a higher energy.

Line 282- "Numerous pits..." Are there more pits in the walnut sclerenchyma as compared to pistachio? Are there more pits in the dense as compared to porous walnut shell? Can you measure the pit density?

Line 288- "influence crack propagation" - you have no evidence for this and therefore this is an interpretation to the data- please move to discussion. The nice LEGO connection is not in place either. I suggest separation of Fig. 4e and the inset in Fig. 4d to a fifth figure, which is hypothetical and presented in the discussion. The sentence at line 289-290 is not clear, and I suspect that it is too hypothetical, and not based on measurements.

Figure 4- Missing parenthesis after ref. [15]. Typo on the title in panel d- missing dense part walnut. This panel is not very clear- I suggest moving the arrowhead to a more central position in the image. As it seems to point to the figure edge. This panel has arrows pointing to the LEGO motifs that may be confused with the arrow pointing to the missing lobe.

Discussion-

In general, the discussion is not focused, and thus the reader is lost. I suggest to concise the text and add a figure explaining your hypotheses regarding the mechanical model, highlighting the difference between the four tissues you analyzed.

Line 319-on- Please move the paragraph to active rather than passive voice.

Line 328- the sentence starting at the end of this line is unclear and too long. In addition, a reference is needed for the comment on "remarkable mechanical performance".

Line 330- seA shells (?)

Line 350- I am not sure about this statement- pits are for cell-cell connections through plasmodesmata, not cell-extracellular space connection.

Line 351- transport of monolignol to the apoplast is possibly done by diffusion, specialized transporters, or exocytosis, but not through pits.

Line 353-4 - please cite a reference for this claim

Reviewer: 2

Comments to the Author(s)

In their manuscript "Twist and Lock nut shell structures for high strength and energy absorption" Xiao et al report the structural, chemical and mechanical characterisation of walnut and pistachio shells and provide novel insight in the underlying structure/chemistry - function relationships.

In general, the manuscript is of high quality, including very interesting analytical approaches and results; however, before suitable for publication the following points should be addressed:

- 1) The authors report a different lignin content for the two different nut samples (based on IR/Raman measurements, semi-quantitative methods, and literature values) is it possible for the authors to provide wet chemistry results as quantitative analysis for the lignin content?
- 2) The authors analyse fracture surfaces (after mechanical testing) to elucidate structural features - how to ensure that the fractured surfaces are representative for the native state structure; for

example in the case of wood cells (cell walls) fractured surface reveal a different structure, compared to sample surfaces prepared by other techniques and the native state;

3) Page 5 line 92-ff: “cell lumina were selected and expanded simultaneously until touching each other along the cell wall” –I have to admit that I cannot entirely follow the process described here – the procedure should be described a bit more in detail.

4) Page 6 line 103: IR spectra: the authors mention that spatial average spectra were utilized; what exactly do the authors mean with that – was a mapping performed? Or multiple IR spectra averaged?

5) AFM – Adhesion images (SI): it would be very helpful if the authors could explain more in detail how the data for the lignin/carbohydrates adhesion values were fitted-even though the authors refer to a previous work; it would be interesting to perform the same analysis with the delignified samples; Do the results change accordingly when the porous pistachio (less lignin) is analyzed?

Typically, the topography is influenced by the orientation of the cellulose fibrils – how is this accounted for?

6) Raman Analysis: the authors report changing lignin content between CW and CML/CC with different ratios between pistachio and walnut; lignin peaks, eg. the utilized bands at 1658 and 1600 1/cm, (intensity) can also be influenced by a different lignin structure/not only amount or by aromatic extractives (common for walnut)-can the authors comment on that? In particular, considering that in the provided images in Figure 2b it looks like there is quite some underlying fluorescence (also reported by the authors in a previous paper for walnut shells+S10); Are the averages taken from different cell walls;

7) Mechanical properties: in the case of the walnut the analyzed sample comprise dense and porous parts, whereas for pistachio dense and porous parts are separated? Is this of influence for the results?

8) Figure 4 d: I assume the right part should be the dense part;

9) Perspectives of using the waste shell waste for materials: can the authors be a bit more specific how the waste could be utilized in composite materials and to profit from the nutshell intrinsic structure.

===PREPARING YOUR MANUSCRIPT===

===PREPARING YOUR REVISION IN SCHOLARONE===

Author's Response to Decision Letter for (RSOS-210399.R0)

See Appendix A.

RSOS-210399.R1 (Revision)

Review form: Reviewer 1

Is the manuscript scientifically sound in its present form?

Yes

Are the interpretations and conclusions justified by the results?

Yes

Is the language acceptable?

Yes

Do you have any ethical concerns with this paper?

No

Have you any concerns about statistical analyses in this paper?

No

Recommendation?

Accept as is

Comments to the Author(s)

The authors revised the MS to my satisfaction

Review form: Reviewer 2

Is the manuscript scientifically sound in its present form?

Yes

Are the interpretations and conclusions justified by the results?

Yes

Is the language acceptable?

Yes

Do you have any ethical concerns with this paper?

No

Have you any concerns about statistical analyses in this paper?

No

Recommendation?

Accept as is

Comments to the Author(s)

The authors addressed the comments in a sufficient way and made respective changes in the manuscript.

Decision letter (RSOS-210399.R1)

Dear Dr Gierlinger,

It is a pleasure to accept your manuscript entitled "Twist and lock: nut shell structures for high strength and energy absorption" in its current form for publication in Royal Society Open Science.

The comments of the reviewer(s) who reviewed your manuscript are included at the foot of this letter.

on behalf of Professor Guy Genin (Associate Editor) and Pietro Cicuta (Subject Editor)
openscience@royalsociety.org

Associate Editor Comments to Author (Professor Guy Genin):

Associate Editor: 1

Comments to the Author:

Congratulations on this excellent paper, and many thanks to you for submitting your best work to Royal Society Open Science.

Reviewer comments to Author:

Reviewer: 1

Comments to the Author(s)

The authors revised the MS to my satisfaction

Reviewer: 2

Comments to the Author(s)

The authors addressed the comments in a sufficient way and made respective changes in the manuscript.

Appendix A

First, we would like to thank both reviewers for their valuable and constructive comments, which helped to improve our manuscript. Below, we have addressed every question and suggestion point by point.

Associate Editor Comments to Author (Professor Guy Genin):

Associate Editor: 1

Comments to the Author:

Thank you for submitting this very interesting paper to RSOS. Like me, the reviewers find your results and ideas compelling. The reviewers have some constructive criticism about how your paper might be improved, and I encourage you to follow their advice in the revised manuscript.

Associate Editor: 2

Comments to the Author:

(There are no comments.)

Reviewer comments to Author:

Reviewer: 1

Comments to the Author(s)

The article presents the structure and mechanical characteristics of nutshells, comparing pistachio and walnut. The interpretation of the mechanical analysis is based on the structural and compositional analysis at the micro- to nano- scales. This innovative work is presented in very clear figures. In some cases, interpretation is entangled to the results that makes it hard to distinguish measurement data from implications. This is especially present when mechanical data is interpreted based on microscopy of fractured surfaces- Fig. 4. A clear separation between collected and inferred data will strengthen this work significantly.

Specific comments:

Abstract- I am not sure to follow the connection between pits and lignification.

Thanks for pointing to this unclear wording. The sentence was rephrased for better understanding. To fulfil the above mentioned “clear separation between collected and inferred data” we now describe that walnut often fracture along the pit channels and not that pit channels are responsible for the brittle behaviour.

Line 72- Please explain why frozen samples are the best to preserve the structure, as freezing may damage the sample.

The sentence is changed for better understanding. Freezing was chosen to minimize chemical changes (e.g. oxidation or microorganism decay). As our focus is on secondary cell wall tissues (sclerenchyma) with a very low water content we can anticipate that freezing will not (or only

very minor) damage the sample structure. We agree that freezing would be critical for primary plant cell wall tissues with high water content.

Line 213- what does the normalization peak at 1374 cm⁻¹ represent?

We added the interpretation about the peak at 1374 cm⁻¹ and a reference.

The peak at 1374 cm⁻¹ in the IR spectra represents C-H deformation in cellulose and hemicelluloses. To compare the lignin intensity in different samples, normalization at this peak will give changes in lignin content in relation to the carbohydrates and cancel out differences in cell wall thickness (=material available for absorption).

Line 214- The peak ratio is not reported- we (readers) do not see the 3X factor.

We added the relative intensity of the lignin band in walnut (0.7) and pistachio (0.2) as suggested.

Line 217- some word rearrangement is needed- either explain the abbreviation or remove parenthesis

We added “compound middle lamella” to explain “CML”.

Line 219- Please add “in the Raman spectra” or something in this line.

We added “in the Raman spectra” as recommended.

Line 254- Figure 3 does not have mechanical information, but only topography, measured by SEM and AFM. Based on topography you suggest a helicoidal structure. Panel c is very confusing because it is presented as if the SEM and TEM images are of similar scales and similar structures, but the TEM features are about 3-5 times smaller. Please refer to this in the text and explain what the (SEM) bigger features are. I suggest moving the TEM panels together with the high resolution SEM in panel d.

Thanks for pointing to this mistake; we removed “mechanical properties” from the figure legend. All mentioned images in figure3 were rearranged according to your suggestion and to avoid the mentioned confusion the SEM images with bigger features were removed - it is not really needed, the zoom in is enough.

Line 268- why DESPITE the lower strength? Low strength may appear together with high or low strain. The high strain in both dense and porous tissues is very interesting and worthy of a separate discussion.

We changed “despite the lower strength” to “with lower strength” for better understanding.

Thank you for this comment. It is indeed interesting to discuss and we added some sentences in the discussion section.

Line 270- another important variation is the increased lignification. Is the higher strength due to wall density or wall lignin content? The place to discuss this is the discussion. Appearing in the results part, this confuses the reader.

The sentence has been deleted and the effect of lignin is now discussed along with other data in the discussion. We have no data for cell wall density, but as lignin fills out pores between cellulose fibrils it could be that lignin increases cell wall density. On the other hand less lignin might simple be the reason behind smaller lamellae and higher pitch angle which in turn again influences mechanical properties. We discuss these issues now in the discussion, which we tried to improve and to make it more focussed.

Line 275- the striking variation between walnut and pistachio is the high strain. This results in a higher energy.

Yes, to calculate the energy absorption of the specimens we have to integrate the area under stress strain curve. Thus, the higher strain in pistachio results in higher energy absorption. This is now also briefly mentioned in the text.

Line 282- “Numerous pits...” Are there more pits in the walnut sclerenchyma as compared to pistachio? Are there more pits in the dense as compared to porous walnut shell? Can you measure the pit density?

Yes, there are definitely more pits in the walnut sclerenchyma as compared to pistachio, as shown in the Raman and TEM images (electronic supplementary material, figure S10) and also the SEM images of fracture surfaces revealed much less pits in pistachio.

Comparing number of pits in porous (younger development) and dense (older tissue) is difficult as they change during development. In young tissue they are more open, but with cell wall thickening they get more and more closed and finally impregnated with extractives in the mature tissue. So they appear different (see Figure below), but probably the number is not changing, but the characteristics (size, branching) and composition.

Figure. Comparison of Raman fluorescence images of dense and porous walnut shell

It is difficult to measure the pits density of sclereids in walnut and pistachio due to the polylobate shape. Pits occur in all secondary cell wall, including the lobes (figure 4d) and can also be branched (electronic supplementary material, figure S10a).

Line 288- "influence crack propagation"- you have no evidence for this and therefore this is an interpretation to the data- please move to discussion. The nice LEGO connection is not in place either. I suggest separation of Fig. 4e and the inset n Fig. 4d to a fifth figure, which is hypothetical and presented in the discussion. The sentence at line 289-290 is not clear, and I suspect that it is too hypothetical, and not based on measurements.

We agree and have deleted the speculative sentence and moved parts to the discussion as suggested.

We appreciated very much the idea of a fifth figure and rearranged as recommended.

We added annotation for arrows including compound middle lamella (CML), lobes and pits for easier understanding.

Figure 4- Missing parenthesis after ref. [15]. Typo on the title in panel d- missing dense part walnut. This panel is not very clear- I suggest moving the arrowhead to a more central position in the image. As it is it seems to point to the figure edge. This panel has arrows pointing to the LEGO motifs that may be confused with the arrow pointing to the missing lobe.

The figure 4 is corrected as suggested.

Discussion-

In general, the discussion is not focused, and thus the reader is lost. I suggest to concise the text and add a figure explaining your hypotheses regarding the mechanical model, highlighting the difference between the four tissues you analyzed.

We improved the discussion, also by adding now the suggested Figure 5 to explain our hypotheses regarding the mechanical model. We worked a lot on the discussion to make it more focussed and think that it has been improved.

Line 319-on- Please move the paragraph to active rather than passive voice.

The sentence is rephrased to active voice as suggested.

Line 328- the sentence starting at the end of this line is unclear and too long. In addition, a reference is needed for the comment on "remarkable mechanical performance".

The sentence is rephrased for better understanding.

Line 330- seA shells (?)

Thank you, we corrected the typo.

Line 350- I am not sure about this statement- pits are for cell-cell connections through plasmodesmata, not cell-extracellular space connection.

We changed accordingly.

Line 351- transport of monolignol to the apoplast is possibly done by diffusion, specialized transporters, or exocytosis, but not through pits.

The sentence is rephrased for better understanding. Following the pits formation, would serve to maximize the symplast/apoplast contact area, which facilitates diffusion into all regions of the cell wall simply by shortening diffusion distances. We only discuss that the higher lignification might be linked with high number of pit channels to ease transport and connection.

Line 353-4 - please cite a reference for this claim

We rephrased the sentence and added a reference.

Reviewer:2

Comments to the Author(s)

In their manuscript "Twist and Lock nut shell structures for high strength and energy absorption"

Xiao et al report the structural, chemical and mechanical characterisation of walnut and pistachio shells and provide novel insight in the underlying structure/chemistry – function relationships. In general, the manuscript is of high quality, including very interesting analytical approaches and results; however, before suitable for publication the following points should be addressed:

1) The authors report a different lignin content for the two different nut samples (based on IR/Raman measurements, semi-quantitative methods, and literature values) is it possible for the authors to provide wet chemistry results as quantitative analysis for the lignin content?

The different chemistry of the two nut samples was proven by two spectroscopic methods (Raman microscopy and FT-IR spectroscopy), histochemistry and confirmed by a very recent study, by Landucci et al., (2020) reporting 17% total lignin in pistachio shells and 32% lignin in walnut shells. As our emphasis was to reveal not only chemistry per species, but also differences between porous and dense as well as between cell wall and interface (CML) our methods of choice were microspectroscopic approaches.

Reference: “Landucci, L, et al., Eudicot Nutshells: Cell-Wall Composition and Biofuel Feedstock Potential. *Energy Fuel* 2020, 34, (12), 16274-16283.”

2) The authors analyse fracture surfaces (after mechanical testing) to elucidate structural features – how to ensure that the fractured surfaces are representative for the native state structure; for example in the case of wood cells (cell walls) fractured surface reveal a different structure, compared to sample surfaces prepared by other techniques and the native state;

In this work, we elucidated the native structural features not only based on the fracture surfaces by SEM, but also by Raman and TEM images, as well as AFM topography. Many of the structural features were consistent regardless method and sample preparation. For example: more pit channels were observed in walnut than pistachio, shown in SEM images of fractured cells (figure 4d-e) as well as in microtome cut cells by Raman and TEM images (electronic supplementary material, figure S10a and b). The ball-joint-like connection in fracture images by SEM (figure 4g) was also confirmed in Raman images (figure 2b).

But, we agree that fracture has an influence on certain features. For example, we observed gaps between the lamella in the SEM images in Figure 3c. Therefore, these images were not used for the thickness calculation.

3) Page 5 line 92-ff: “cell lumina were selected and expanded simultaneously until touching each other along the cell wall” –I have to admit that I cannot entirely follow the process described here – the procedure should be described a bit more in detail.

Thanks for pointing to this unclear description. We describe now in more detail the 3D reconstruction.

4) Page 6 line 103: IR spectra: the authors mention that spatial average spectra were utilized; what exactly do the authors mean with that—was a mapping performed? Or multiple IR spectra averaged?

We improved the whole paragraph to have all the information included and rephrased for better understanding.

We obtained multiple IR spectra from specific regions of interest by mapping using an IR transmission microscope. For analysis these spectra (for pistachio dense: 90, pistachio porous: 60; walnut dense: 90; walnut porous: 50) were averaged for every region of interest.

5) AFM – Adhesion images (SI): it would be very helpful if the authors could explain more in detail how the data for the lignin/carbohydrates adhesion values were fitted—even though the authors refer to a previous work; it would be interesting to perform the same analysis with the delignified samples; Do the results change accordingly when the porous pistachio (less lignin) is analyzed? Typically, the topography is influenced by the orientation of the cellulose fibrils – how is this accounted for?

We have not analysed delignified samples and porous pistaccio as the sample preparation has not been successful yet. But we have AFM measurements and thus adhesion values of pure lignin by extracting values from the middle lamella of an AFM measurement on pistachio (supplementary material, figure S4). The adhesion values derived from the middle lamella only result in a sharp distribution of low adhesion values. As the middle lamella is known to contain mainly lignin (Fig. 2, Raman data), the assignment of lignin to the low adhesion values can be confirmed.

We agree that one should see the orientation in the AFM topography images, but the resolution (AFM tip 10nm) is unfortunately not high enough to reveal the orientation within single lamella. Another problem is also that cellulose fibrils with different angles appear elongated after cutting. Therefore, we show the AFM stiffness values in the supplementary material (Figure S5), which clearly show different values for different orientation.

6) Raman Analysis: the authors report changing lignin content between CW and CML/CC with different ratios between pistachio and walnut; lignin peaks, eg. the utilized bands at 1658 and 1600 1/cm, (intensity) can also be influenced by a different lignin structure/not only amount or by aromatic extractives (common for walnut)-can the authors comment on that? In particular, considering that in the provided images in Figure 2b it looks like there is quite some underlying fluorescence (also reported by the authors in a previous paper for walnut shells+S10); Are the averages taken from different cell walls;

The bands at 1658 and 1600 cm^{-1} represent C=C/C=O and aromatic ring stretches, respectively. In lignin, they originate mainly from ethylenic residues (Agarwal and Ralph 2008, Determination of ethylenic residues in wood and TMP of spruce by FT-Raman spectroscopy, *Holzforschung* 62:667 - 675DOI: 10.1515/HF.2008.112, Bock and Gierlinger 2019 Infrared and Raman spectra of

lignin substructures: Coniferyl alcohol, abietin, and coniferyl aldehyde JRS 50, 6 <https://doi.org/10.1002/jrs.5588>). So other lignin structures can only contribute to these bands insofar as they contain C=C/C=O groups in conjugation with aromatic rings. In lignins, these are only a few structures, namely cinnamyl alcohols, their aldehydes and acids. If the lignin is differing in other structural aspects (e.g. amount of β - β motifs, different S:G ratio), it is not reflected in these bands.

The same is true also for extractives. Only extractives bearing either C=C/C=O or aromatic ring structures can have bands at this position. In walnut, naphthoquinones (juglone etc) indeed have a strong band at 1660 cm^{-1} . However, the experimental spectra show less intensity of this band in walnut than in pistachio. Hence, we conclude that naphthoquinones do not play a role in the spectrum and that can be used for lignin interpretation.

Lastly, we also tried washing samples with ethanol, acetone etc. and we could not observe any spectral differences to the native one. Washing should remove such structures and therefore we also conclude here, that extractives do not play a role in the spectra.

Finally, the highly fluorescing component were found in the pits of the dense walnut part and inner layer of cell wall in porous walnut tissue. Extraction with solvents of the sections did not remove the fluorescence background, which points to a non-extractable phenolic component or lignin.

And anyway, the fluorescence background will influence both bands used for the ratio (1658 and 1600 $1/\text{cmintensity}$).

Yes, the spectra shown in the Figure S3 were the average spectra taken from different cell walls.

7) Mechanical properties: in the case of the walnut the analyzed sample comprise dense and porous parts, whereas for pistachio dense and porous parts are separated? Is this of influence for the results?

Walnut shells at the mature stage show mainly dense parts and only the very inner part is porous (Fig. 1c). After trimming the samples for mechanical testing the porous part is almost away and thus we expect no/minor influence on the mechanics. On the other hand the porous part is too small to be separated for mechanical testing and also in young state walnut tissue show a strong gradient (from thick walled to thick walled and also in terms of lignification, see Xiao et al 2019). But fortunately this is not the case in pistaccio: the mature pistachio shells are just composed of dense tissue (Fig. 1b), the young pistaccio shells only of porous tissue (Fig. 1a). With this clear and uniform separation an idea on the influence of density (cell wall thickness) can be gained.

8) Figure 4 d: I assume the right part should be the dense part;

Thank you very much for reporting this mistake in Figure 4d. We changed “porous part” to “dense part”.

9) Perspectives of using the waste shell waste for materials: can the authors be a bit more specific how the waste could be utilized in composite materials and to profit from the nutshell intrinsic structure.

We added perspectives in future work as recommended